# CO$_2$ Microwave Plasma—Catalytic Reactor for Efficient Reforming of Methane to Syngas

**Se Min Chun** [1] , **Dong Hun Shin** [1] , **Suk Hwal Ma** [1,2] , **Geon Woo Yang** [1,3]
**and Yong Cheol Hong** [1,*]

1 Plasma Technology Research Center, National Fusion Research Institute, 37 Dongjansan-ro, Gunsan, Jeollabuk-do 54004, Korea; semin83@nfri.re.kr (S.M.C.); shindong17@nfri.re.kr (D.H.S.); shma@nfri.re.kr (S.H.M.); gwyang8@nfri.re.kr (G.W.Y.)

2 Department of Applied Plasma Engineering, Electronic Engineering, Chonbuk National University, 567 Baekje-daero, Deojin-gu, Jeonju, Jeollabuk-do 54896, Korea

3 Department of Bio Nano System Engineering, Division of Mechanical Design Engineering, Chonbuk National University, 567 Baekje-daero, Deojin-gu, Jeonju, Jeollabuk-do 54896, Korea

* Correspondence: ychong@nfri.re.kr; Tel.: +82-440-4111

**Abstract:** CO$_2$ was converted to synthesis gas in a microwave plasma–catalytic reactor by methane reforming at atmospheric pressure. The hybrid system used waste heat from the plasma to heat the catalyst. Conversion degrees were examined as a function of gas temperature, and the reforming efficiency of the plasma-only system was compared with that of the hybrid system. As a result, the hybrid system was shown to be more efficient under catalyst-free conditions. The use of microwave plasma alone resulted in low conversions of CO$_2$ and CH$_4$, which were 32.9% and 42.7%, respectively, at 3 kW microwave power. High CO$_2$ and CH$_4$ conversions of 69.3% and 81.2%, respectively, were achieved in the presence of catalyst at the same microwave power. At constant microwave power, catalyst addition increased the H$_2$ and CO mass yield rates to 0.177 kg/h and 1.58 kg/h, respectively. Additionally, the H$_2$ energy yield were 59.1 g/kWh. Thus, the developed hybrid system is well suited for efficient and economically viable CO$_2$ reduction and synthesis gas production, paving the way for next-generation CO$_2$ utilization and zero-emission industrial processes.

**Keywords:** carbon dioxide; reforming; microwave plasma; catalyst; synthesis gas

## 1. Introduction

The ever-increasing levels of atmospheric carbon dioxide (CO$_2$) significantly contribute to global climate change, necessitating the search for methods of reducing the emissions of this greenhouse gas to zero. The international community has made various efforts to reduce greenhouse gas emissions since the adoption of the Climate Change Convention in 1992 (the Kyoto protocol). The Kyoto protocol mainly aimed to reduce greenhouse gas emissions in advanced countries, but it was not properly implemented. At the Paris climate conference (COP21) held in December 2015, 195 countries agreed on the first ever universal and legally binding global climate deal. This agreement sets a long-term goal of keeping the global average temperature increase to well under 2 °C above pre-industrial levels and foresees the undertaking of rapid reductions in accordance with the best available methods. Advanced countries have therefore established long-term targets for reducing greenhouse gas levels and are currently introducing greenhouse gas abatement technologies such as CO$_2$ capture and storage (CCS) [1,2], CO$_2$ capture and utilization (CCU) [3,4], and CO$_2$ reforming of methane (CH$_4$) [5]. However, despite being the primary technology used in industry for CO$_2$ capture, CCS is not a fundamental solution. It is inferior to methods that convert CO$_2$ into energy sources or other useful

substances. Reforming technology utilized for CCU is more attractive, because it not only reduces $CO_2$ emissions while consuming $CH_4$, but it also affords a synthetically useful mixture of hydrogen ($H_2$) and carbon monoxide (CO). Widely used reforming technology employs plasma or a catalyst for a dry reforming reaction with $CH_4$ or a similar hydrocarbon material and $CO_2$ to produce a synthesis gas (Syngas) containing $H_2$ or CO, thereby recycling $CO_2$. However, the dry reforming process is a highly endothermic and requires special methods for achieving a considerable reaction rates to satisfy industrial requirements. Plasma and catalyst-based techniques have not been commercialized, even though they are regarded as having the potential to satisfy the conditions required by industry. Reforming a large amount of $CO_2$ or $CH_4$ by a conventional dry reforming method using plasma has many limitations. Standards for manufacturing plasma reformers, where plasma reforming reactions occur, are restrictive. The efficiency of plasma reforming is lower than reforming methods that employ a catalyst. Furthermore, the system used for conventional dry reforming with a catalyst is complicated, since it requires strict control of heat to maintain a suitable temperature to activate the catalyst in consideration of an endothermic reaction. If heat is improperly controlled, the catalyst may perform poorly. The catalysts can deteriorate at high temperatures, and it may adsorb a considerable amount of carbon.

In this article, a hybrid system was conceived to solve these. One aspect of the hybrid system is the use of $CO_2$ plasma and a catalyst, which can maximize reforming efficiency, since the supplied $CH_4$ gas primarily undergoes a dry reforming reaction with $CO_2$ microwave plasma. A dry reforming reaction with the catalyst. We present a $CO_2$ torch that makes use of microwaves without electrodes and investigate the dissociation properties of $CO_2$ molecules in a high temperature torch. The $CO_2$ microwave plasma torch is very simple, so our device is compact. The $CO_2$ microwave plasma torch may contain highly active radicals, particularly oxygen atoms, which serve to increase the chemical reaction rate. Additionally, the high temperature of the microwave plasma flame supplies heat to the reactor wall between the plasma reaction zone and the catalytic reaction zone. Therefore, the efficiency of the reforming reaction can be maximized with the waste heat of the plasma. Spectroscopic diagnostics indicated that the carbon dry reforming (CDR) reaction was induced at high temperature and that active radicals, oxygen species, were produced in the $CO_2$ microwave plasma. The production of syngas through dry reforming confirmed that the developed hybrid system can be used to effectively achieve methane conversion at atmospheric pressure.

## 2. Result and Discussion

### 2.1. Analysis of $CO_2$ Dissociation in Torch

$CO_2$ molecules passed through an extremely high temperature torch, where a local thermodynamic equilibrium (LTE) was assumed at T > 2000 K. Almost all energy discharged in these experiments was associated with the dissociation process, syngas production, and numerous chemical and relaxation processes taking place simultaneously in the $CO_2$ plasma. This is important in both fundamental and applied respects [6].

The $CO_2$ dissociation process was assumed to start with and be limited by $CO_2$ dissociation into CO and atomic O, shown in (Equation (1)):

$$CO_2 \rightarrow CO + O \, , \, \Delta H = 5.5 \frac{eV}{mol} \tag{1}$$

The most critical issue was the maximizing energy efficiency of the reaction shown in Equation (1). In such systems, dissociation is mostly controlled by the electronic excitation of $CO_2$, which is not a very energy-efficient process. The energy efficiency of $CO_2$ dissociation in glow discharge experiments does not usually exceed 8%, whereas electron cyclotron resonance and radiofrequency discharge under non-equilibrium conditions yield energy efficiencies of 60%. However, the highest energy efficiency of $CO_2$ dissociation (90%) realized so far has been in non-equilibrium microwave plasma at

atmospheric pressure [7]. Therefore, the highest efficiency can be obtained by decomposing $CO_2$ using microwave plasma.

On the other hand, the enthalpy and entropy changes for the reaction represented by Equation (1) were calculated as $\Delta H$ = 530 kJ/(mol K) and $\Delta S$ = 147 kJ/(mol K) from readily available data, and the corresponding Gibbs free energy was determined using the standard relationship:

$$\Delta G = \Delta H - T\Delta S \qquad (2)$$

The temperature of $CO_2$ decomposition into CO and O was calculated as $T \approx 3600$ K, and this theoretically obtained value was compared with that predicted by simulation software (Chemical WorkBench, Kintech Lab, Moscow, Russia). In the latter case, the onset of $CO_2$ dissociation was predicted to occur at ~2400 K. At 3600 K, half of the molecules dissociated, and almost complete dissociation was achieved at 7000 K [8]. Since the dissociation of $CO_2$ occurs at high temperature, it is commonly achieved using numerous plasma systems. Atmospheric-pressure microwave plasma has a wide temperature distribution, with the mean temperature exceeding 6500 K [9,10]. Moreover, the temperature-dependent speciation of $CO_2$ microwave plasma is well described in the range of 2000 K $< T <$ 7000 K [8]. Herein, $CO_2$ dissociation was achieved by means of electronic excitation in microwave plasma, which had a sufficiently high temperature to achieve suitable energy efficiency. The dominant conversion of $CO_2$ and $CH_4$ proceeded according to Equation (3):

$$CO_2 + CH_4 \rightarrow 2CO + 2H_2 \qquad (3)$$

The corresponding enthalpy and entropy changes were calculated as $\Delta H$ = 247 kJ mol$^{-1}$ and $\Delta S$ = 257 kJ/(mol K) [11]. The onset temperature of the reforming process was calculated to be $T = \Delta H/\Delta S$ = 961 K. Notably, the temperature of the $CO_2$ microwave plasma is much higher than this value, which is easy to achieve in most plasma flames at atmospheric pressure. Furthermore, $CO_2$ microwave plasma is known to contain highly active species, such as electrons, ions, and radicals, which enhance the reaction rate and eliminate the need for catalysts during materials processing. In particular, atomic oxygen is highly reactive and plays an important role in dry reforming. The results obtained suggested that the introduction of hydrocarbons into the $CO_2$ microwave plasma torch would result in their conversion into new materials, while the $CO_2$ concentration in the torch would be concomitantly reduced.

## 2.2. Temperature and Speciation Measurement of CO$_2$ Microwave Plasma Torch Flame

Generally, gas temperatures are inferred from the rotational temperatures. However, the bulk gas temperature directly affects the kinetics of the chemical reactions in the plasma torch. Therefore, study of gas temperatures in the plasma torch is important for syngas production, gas dissociation, and other applications. Although several methods can be used to determine the gas temperature, optical emission spectroscopy (OES) is commonly used, because it is simple and can be used in situ [9]. Also, OES is an important method for plasma diagnostics. This is because the kinetic energy of electrons in the plasma is partially converted into the internal energy of the molecules present, thus inducing various electronic transitions and resulting in the emission of light. Hence, analysis of emitted light allows the determination of the distribution and internal states of species within the plasma. Figure 1 shows a typical $CO_2$ microwave plasma flame generated at an applied power of 3 kW and a $CO_2$ gas flow rate of 15 L min$^{-1}$. In our preliminary experiments, we observed that the above plasma was stabilized at a flow of 5 L min$^{-1}$ kW$^{-1}$, where it displayed a longer column and larger diameter. The flame volume was almost linearly proportional to the microwave power. Emission analysis was performed at 15-mm intervals from the upper wall of the tapered waveguide. A collimating lens was mounted on each cylindrical emission hole at the dividing position. The $CO_2$ microwave plasma flame comprised two distinct regions, namely bright whitish and bluish ones. Figure 2 displays optical emission spectra of a typical $CO_2$ plasma flame corresponding to Figure 1. The relative quantities

of the chemical species in the generated plasma were determined at different axial positions of the plasma column, with the dominant products of $CO_2$ decomposition identified as C, O, CO, $C_2$, and $O_2$. The bright region corresponded to the typical high-temperature area of $CO_2$ dissociation, whereas the bluish region represented the recombination zone. The emissions from the $CO_2$ microwave plasma were dominated by that of diatomic carbon at 400–600 nm, atomic oxygen at 777, 844, and 925 nm, atomic carbon, carbon radicals, and $CO_2$ molecules in the UV region. Notably, a broad molecular spectrum principally corresponding to molecular $CO_2$ and CO was observed at increasing distances from the bright plasma region. Therefore, effective $CH_4$ conversion required the utilization of abundant oxygen species, radicals, and other reactive species in the bright regions of the microwave plasma. Specifically, active atomic oxygen in the bright region played an important role in the dry reforming reaction, because it readily reacted with methane to produce CO and $H_2$. In most plasma torch applications, OES is used to find the molecular or rational lines for estimating the rotational/gas temperature of nitrogen or air plasma. The rotational lines of the first negative $B^2\Sigma_\mu^+ \rightarrow X^2\Sigma_g^+$ system of $N_2^+$, particularly those of the (0–0) band, have often been used to determine rotational and vibrational temperatures in plasma torches. The accuracy of the technique is directly related to the number of rotational lines that can be used with global fitting or Boltzmann-plot techniques [12–16]. Herein, the temperature of the $CO_2$ microwave plasma was estimated from the $C_2$ Swan band transition (517–565 nm) related to the rotational structures of diatomic gases, which provided information regarding rotational temperature. The rotational and vibrational temperature in plasmas are often determined by comparing experimental and simulated optical spectra. Figure 2c shows the optical emission spectra of the $C_2$ Swan band in a $CO_2$ microwave plasma analyzed using SPECAIR version 3.0 software. The database includes electronic, vibrational, rotational, and translational excitations in a wide temperature range [17], allowing SPECAIR to calculate simulated spectra that can be compared to experimental results, with the corresponding agreement for the $C_2$ Swan band shown in Figure 2c. This revealed that the experimental high-temperature emission profile was in reasonably good agreement with the simulated one. Based on data obtained at the central axis of the 3-kW plasma flame, the flame temperature $T$ was determined as 6760 K, which was high enough to perform CDR reactions without the need for catalysts or an additional heat source. In short, we expected the reforming reactions of $CO_2$ and $CH_4$ to readily occur in the bright region of the $CO_2$ microwave plasma owing to the presence of sufficient amounts of oxygen radicals and high temperatures.

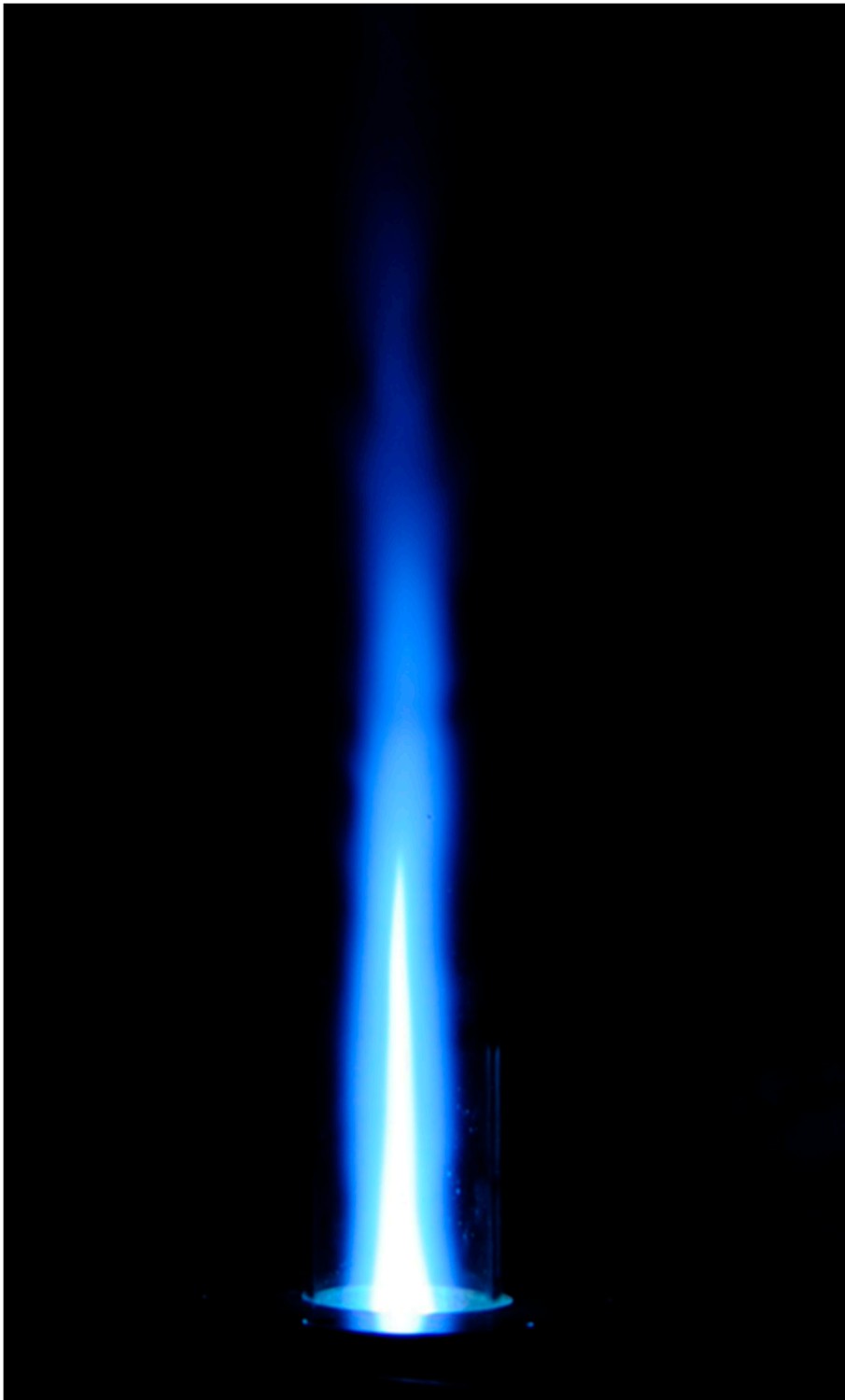

**Figure 1.** Photographs of the $CO_2$ microwave torch plasma within a quartz tube at a microwave operating at 3 kW. The internal diameter and length of the quartz tube were 26 mm, length = 300 mm, respectively.

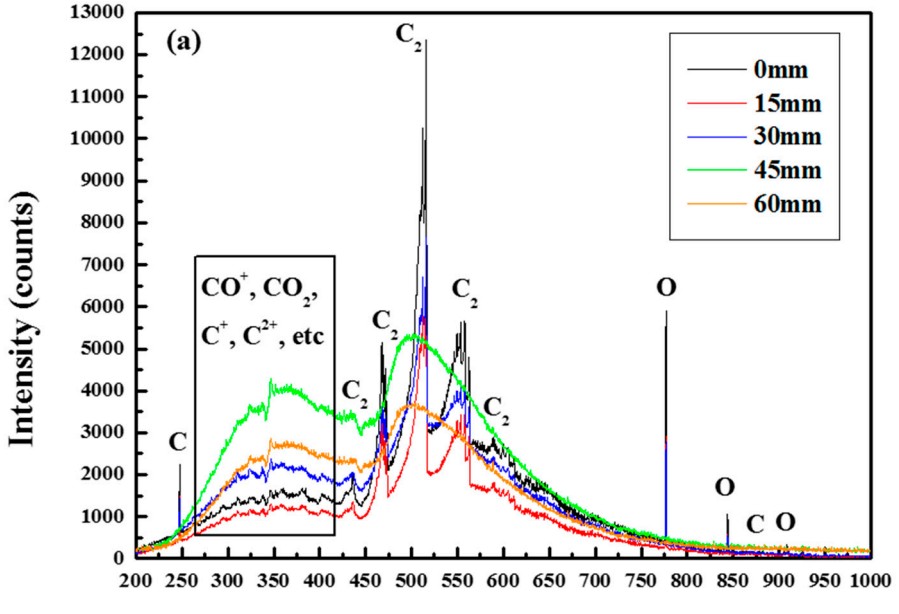

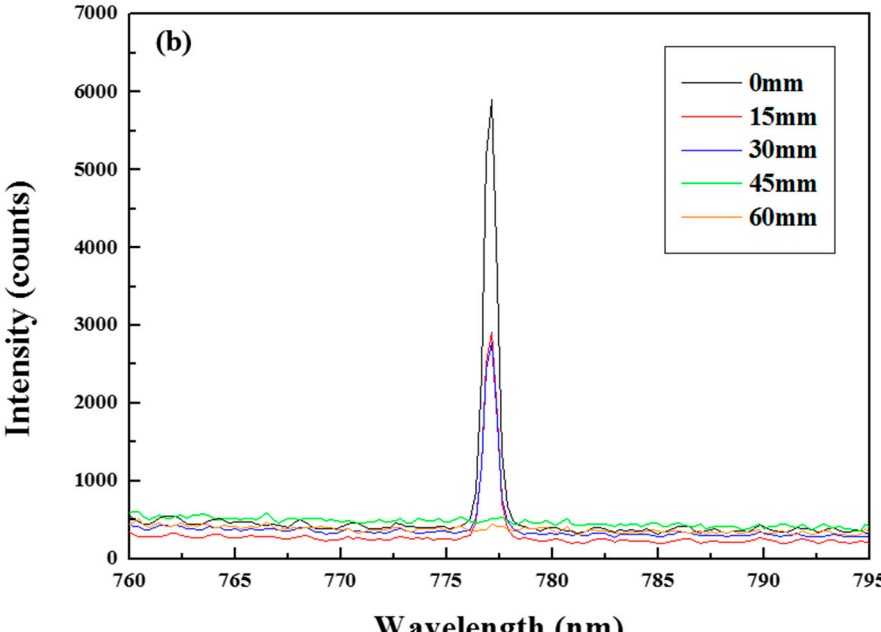

**Figure 2.** *Cont.*

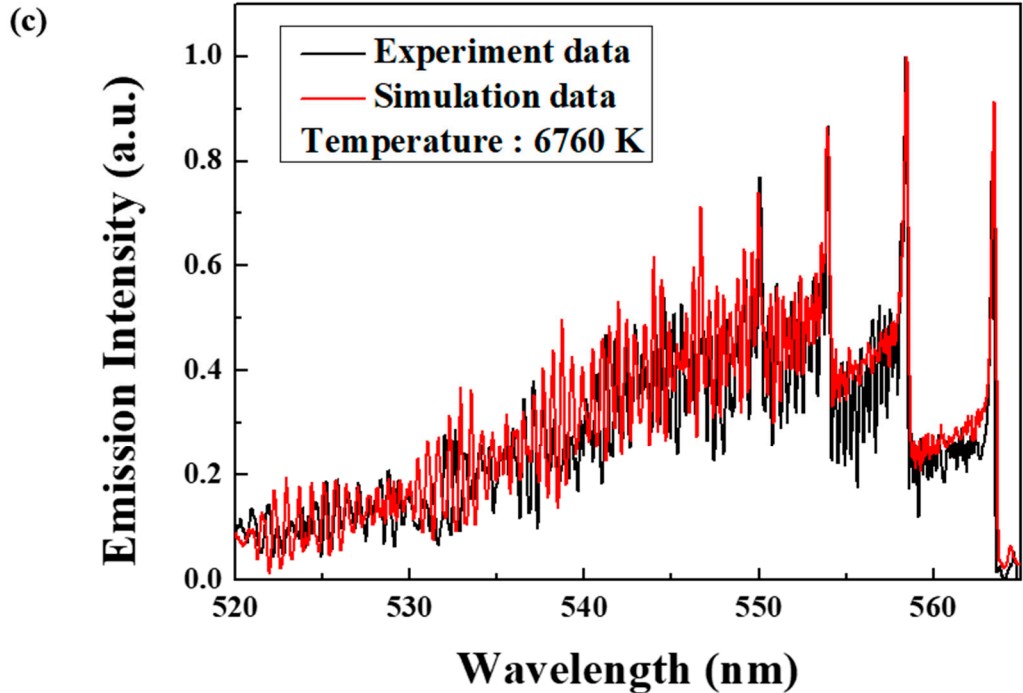

**Figure 2.** Axial position–dependent emission spectra of (**a**) species and (**b**) atomic oxygen generated within the $CO_2$ microwave plasma torch. (**c**) Calculated and measured emission spectra for the $C_2$ Swan band transition in the $CO_2$ microwave plasma torch.

### 2.3. Catalyst Preparation and Utilization

The utilization of catalysts can increase the yield of the reforming process considerably. The catalyst in the microwave plasma–catalytic reactor maintained at 650 °C to increase its activity. From an economic point of view, the use of plasma torch-only systems for reforming is not economically viable. The energy used for heating must be supplied by other processes. Moreover, if the waste heat generated in the plasma torch is not recycled, it results in significant heat dissipation to the surroundings. Thus, the energy used to heat the catalyst can be supplied by plasma waste heat. To optimize the thermal effect of the plasma, the outside of the microwave plasma–catalytic reactor was insulated as shown in Figure 3a, and the temperature of the inner component and outer components were measured over one hour. As shown in Figure 3a, R-type thermocouples were installed at 150-mm intervals in the inner component of the microwave plasma-catalyst reactor, and an additional thermocouple was attached to the outer component of the reactor, with the temperature change measured until saturation. The temperature of the inner component quickly increased after plasma discharge, saturating at ~650 °C, whereas the rear temperature equaled ~550 °C. The outer component temperature of the reactor slowly rose after plasma discharge, saturating at ~600 °C. Although both the innermost and outer component temperatures were slightly lower than the optimal catalytic activity temperature of 800 °C, it was still possible to induce the catalytic reaction using plasma waste heat.

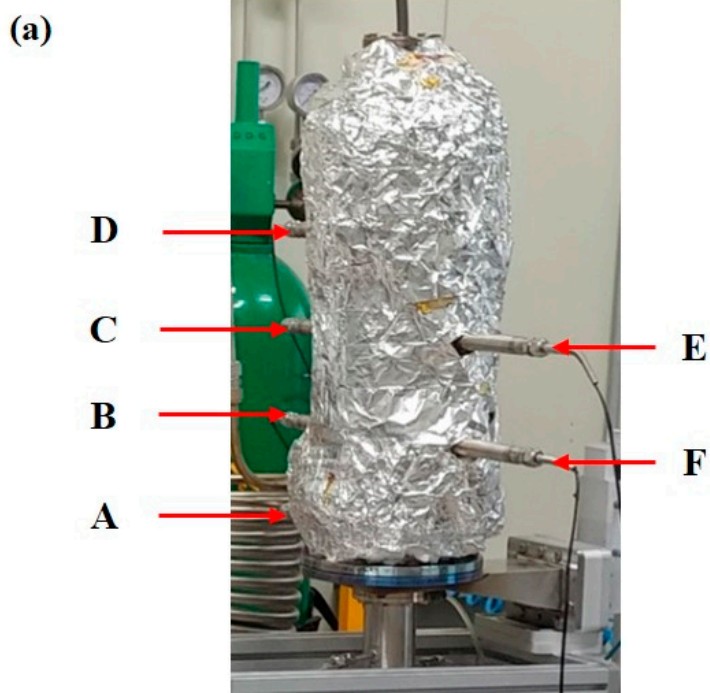

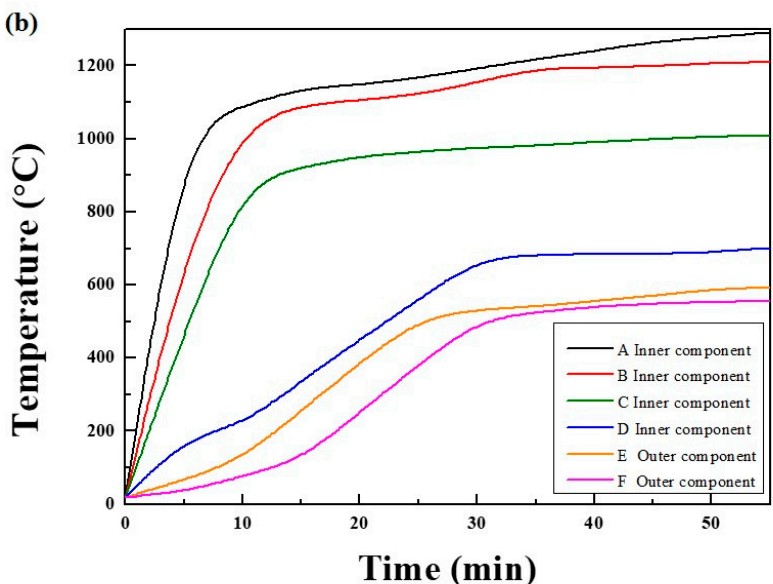

**Figure 3.** (**a**) Photograph of the insulated outer surface of the microwave plasma–catalytic reactor. (**b**) Temperature changes in the inner and outer components measured over 1 h using various thermocouples.

The structure characteristics of the fresh and spent catalysts were analyzed by the adsorption–desorption method and X-ray diffraction (XRD) because the carbon deposition after the reforming reaction caused the deactivation of the catalyst. In order to determine the catalyst surface area and average pore size diameter for the fresh and spent catalysts, the catalysts were analyzed by the adsorption-desorption method. The measurements were carried out with approximately 0.7 g of crushed catalyst sample using a Micromeritics ASAP 2420 multi-port surface area and porosimetry analyzer (Micromeritics Instrument Corp., Norcross, GA, USA). The inset in Figure 4 shows the crushed catalyst samples. The spent catalyst exhibit a darker color than the fresh

catalyst. Therefore, it can be visually confirmed that the spent catalyst contains carbon-containing species. The adsorption-desorption isotherm graph for the fresh and spent catalysts are compared in Figure 4. Increase of adsorption with increasing relative pressure can be noticed at $0.75 < P/P_0 < 0.95$. Even though both catalysts show similar isotherm shapes in the adsorption-desorption curve, the fresh catalysts show an integrated area larger than that of the spent catalysts, implying that the performance of spent catalysts decreased. This can be confirmed from the measurements of specific surface area (SSA). In this context, the specific surface area of catalysts were determined by a Brunauer–Emmett–Teller (BET) measurement. Compared to fresh catalysts (SSA = 7.83 $m^2/g$), the spent catalysts showed a lower surface area (SSA = 4.58 $m^2/g$), resulting from the deposition of carbon powders on the catalyst surface. In the BET measurement, therefore, carbon powder causes a blockage of the pores of catalysts and thus reduces the volume of nitrogen adsorbed. Additionally, the XRD patterns were obtained by an X'pert-pro 40 kW diffraction system with Cu Kα radiation ($\lambda = 1.540598$ Å) at a scan rate of $0.03°2\theta^{-1}$ $s^{-1}$ to confirm whether carbon powders were detected with the spent catalysts or not. The crystal phase was also identified for the fresh and spent catalysts as shown in Figure 5. The fresh and spent catalysts only exhibited characteristic Bragg diffraction peaks can NiO (JCPDS # 01-078-0643) and $CaAl_4O_7$ (JCPDS # 00-023-1037. No clear evidence of traces of carbon was identified in the XRD data. However, weak amorphous carbon peaks could overlap with those of NiO and $CaAl_4O_7$; the broad distribution around 20° may belong to amorphous carbon [18]. This is consistent with carbon deposition during the reforming reaction deactivating the catalyst.

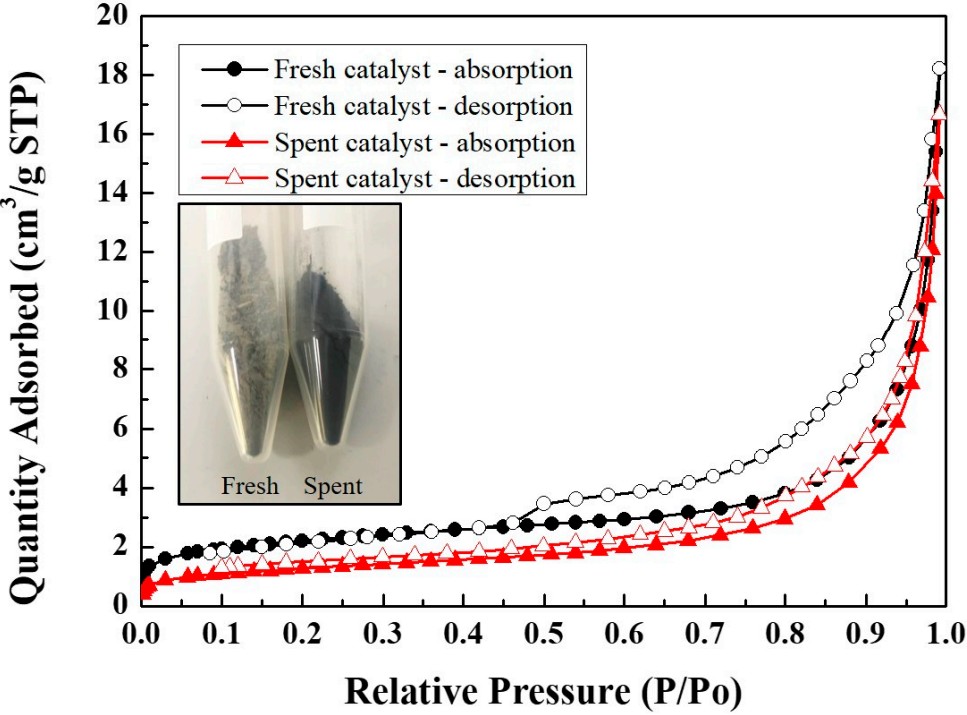

**Figure 4.** Adsorption-desorption isotherms of $N_2$ at 77K for the fresh and spent catalysts.

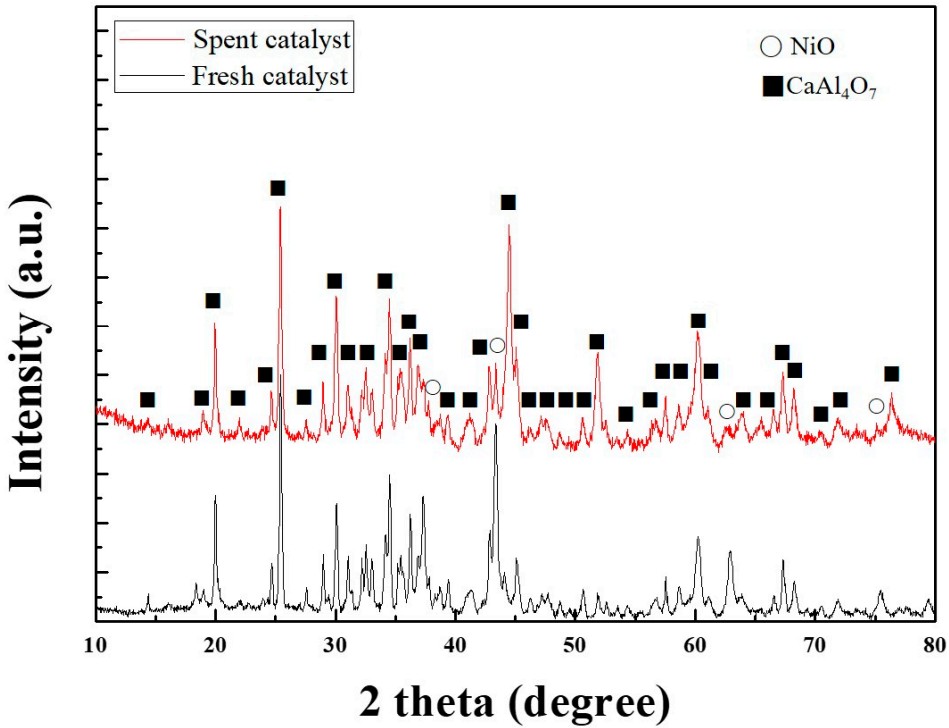

**Figure 5.** XRD patterns of the fresh and spent catalysts.

*2.4. CO$_2$ Microwave Plasma-Catalytic Reactor Performance Evaluation*

Analysis of gaseous products revealed that the produced syngas contained $CO_2$, $CH_4$, $H_2$, and CO. To evaluate the performance of the plasma torch reaction, we determined the syngas concentrations, syngas flow rates, and conversion of $CO_2$ and $CH_4$. The conversions were considered good quantitative indicators of syngas production. To produce $H_2$, hydrocarbon molecules must be cracked, which involves C–C and C–H bond cleavage. Therefore, the cracking efficiency was evaluated using the ratio of atomic carbon contained in the reformed products to carbon contained in the injected hydrocarbons. The conversion of $CO_2$ and $CH_4$ are defined as:

$$CO_2\text{conversion (\%)} = \frac{(CO_2\text{input (mol/s)} - CO_2\text{output (mol/s)})}{CO_2\text{input (mol/s)}} \times 100 \qquad (4)$$

$$CH_4\text{conversion (\%)} = \frac{(CH_4\text{input (mol/s)} - CH_4\text{output (mol/s)})}{CH_4\text{input (mol/s)}} \times 100 \qquad (5)$$

The concentration (a), flow rates (b), and conversions (c) for each experiment are shown in Figure 6. The total gas flow rate in the experiment was 30 L min$^{-1}$, which consisted of 15 L min$^{-1}$ $CO_2$ and 15 L min$^{-1}$ $CH_4$. However, the amount of syngas generated after addition of the catalyst was 1.746 times greater than the quantity of reactants, based on the flow rates. Because reformation of $CH_4$ with $CO_2$ is not equimolar, it generates twice as many moles of product than are supplied by the reactants. Figure 6b shows that addition of the catalyst increased the $H_2$ and CO flow rates from 11.91 to 16.54 L min$^{-1}$ and from 10.71 to 21.1 L min$^{-1}$, respectively. Therefore, these results shown in Figure 6b were applied in Equations (4) and (5) to calculate conversion of $CO_2$ and $CH_4$. On the other hand, gas conversion achieved with the single microwave plasma torch was much lower, as shown in Figure 6c. The conversion of $CO_2$ was very low microwave power of 3 kW. Since the high temperature region of the plasma at low power was not sufficiently larger than that of the plasma at high power, the CDR conditions at low plasma power allowed the CO + O → $CO_2$ recombination reaction to occur, with lower $CO_2$ conversion observed as a consequence. In other words, oxygen and ions could not efficiently react with methane species at low power and were reduced back to $CO_2$ by reverse

reactions with C or CO. The conversion of $CO_2$ in a single microwave plasma torch using high power has been reported elsewhere, and values of 71.3% and 96.8% were achieved at 6 kW for $CO_2$ and $CH_4$, respectively [19]. Therefore, reforming conversion strongly depends on the microwave plasma power. However, as shown in Figure 6, the effect of catalyst addition on conversion in the plasma torch was very strong. The concentration of $CO_2$ was reduced from 23.4% to 9.8%, and that of $CH_4$ was reduced from 20% to 6%. Concurrently, the concentration of $H_2$ increased from 27.7% to 35.2%, while that of CO increased from 24.9% to 44.9%. Figure 6b shows that catalyst addition decreased the flow rates of $CO_2$ and $CH_4$ from 10.06 to 4.61 L min$^{-1}$ and from 8.6 to 2.82 L min$^{-1}$. The flow rate of $H_2$ increased from 11.91 to 16.54 L min$^{-1}$, and that of CO increased from 10.71 to 21.1 L min$^{-1}$. Finally, Figure 6c shows that upon catalyst addition, the conversion of $CO_2$ increased from 32.9% to 69.3%, while that of $CH_4$ increased from 42.7% to 81.2%. These results showed that the conversion efficiency of the hybrid system at 3 kW exceeded that of the plasma-only reactor using high power [19]. Additionally, as shown in Figure 7, the $H_2$ and CO mass yield rates in the hybrid system increased to 0.177 kg h$^{-1}$ and 1.58 kg h$^{-1}$, respectively, which was ascribed to the concomitantly increased conversion of $CO_2$ and $CH_4$. However, from an economic point of view, it was necessary to demonstrate the energy efficiency of the developed hybrid system. To enable comparisons from an economic perspective, the efficiency of the hybrid system was calculated by:

$$\eta = \frac{(H_2)\text{produced} \times \text{LHV}(H_2) + (CO)\text{produced} \times \text{LHV}(CO)}{\text{Input plasma energy} + \text{fuel}(CH_4)\text{ injected} \times \text{LHV}(CH_4)} \tag{6}$$

During the reforming process, hydrocarbons react with oxygen to produce hydrogen, for which the heating value exceeds that of any other hydrocarbon. Therefore, the efficiency of a reforming system can be determined by dividing the lower heating value (LHV) of the produced syngas by the input energy, i.e., the combined electrical energy of the plasma and the LHV of the injected $CH_4$ [20].

It should be noted that this study focused on the efficiency of the plasma reactor, not the overall system. Therefore, this calculation did not take into consideration the energy spent in other process. The obtained results (Figure 8) showed that with identical energy inputs and reactant flow rates, the efficiency of dry reforming increased from 36.8% to 62.1% upon catalyst addition, thus revealing the superior performance of the hybrid system.

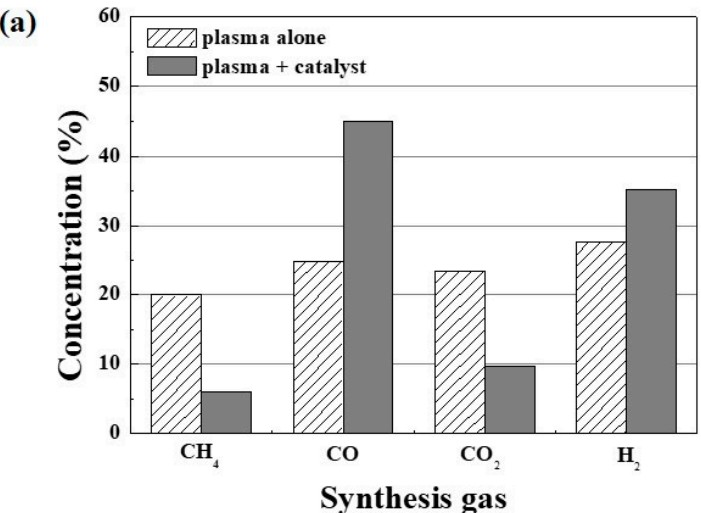

**Figure 6.** *Cont.*

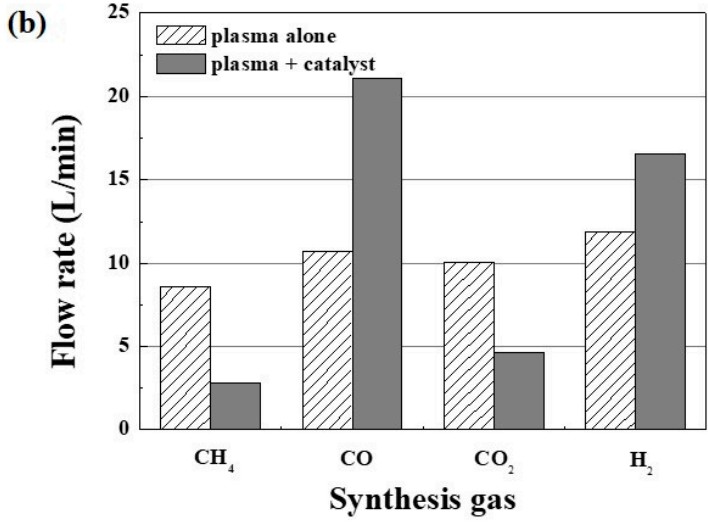

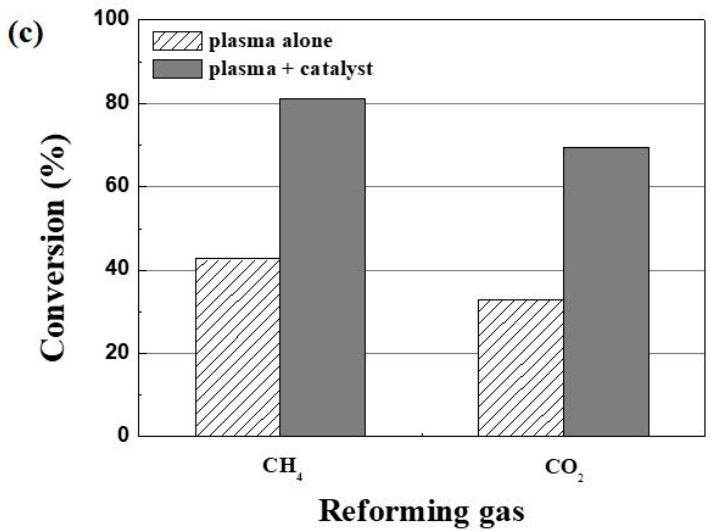

**Figure 6.** (**a**) Concentration, (**b**) flow rate, and (**c**) conversion ratio obtained with different reforming methods.

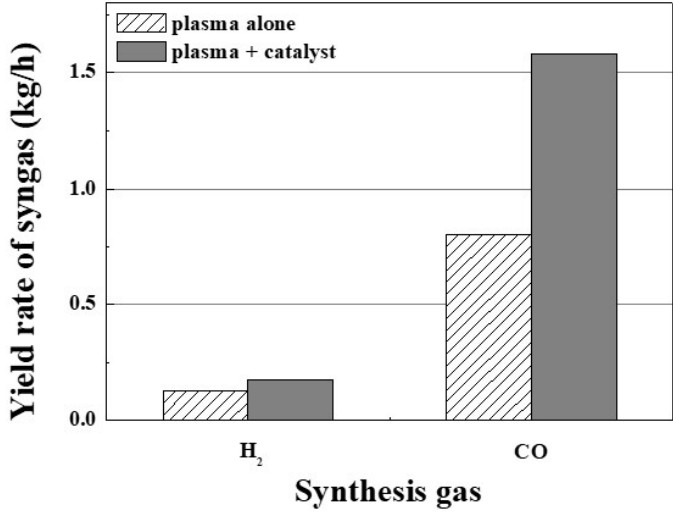

**Figure 7.** Syngas yield rates obtained using the hybrid system.

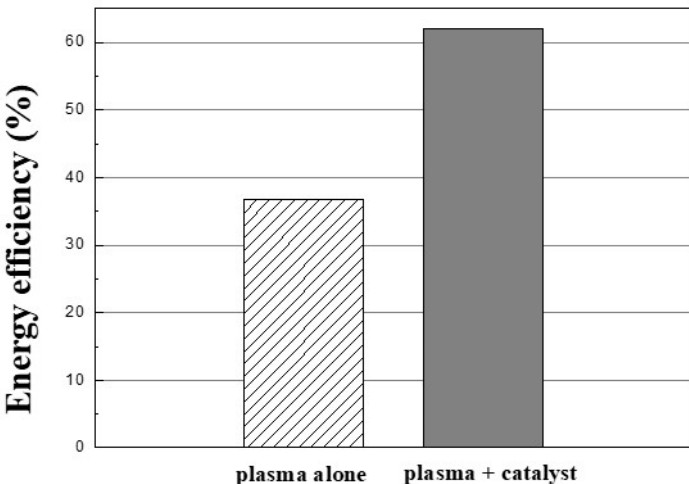

**Figure 8.** Energy efficiency of dry reforming obtained using the hybrid system.

The production rates and energy yields of the various plasma-based method used to produce $H_2$ are compared in Table 1. In the Table 1, fuel for $H_2$ production is classified as either a liquid or a gas. When the two forms of fuel are compared, two differences are obvious. With plasma-based technologies, $H_2$ production is always low when $H_2O$ is present in the fuel. The use of water reduces the plasma efficiency, because it induces an endothermic response. For example, the energy yield of dielectric barrier discharge (DBD) with liquid fuel without $H_2O$ is 0.0067 kg/kWh. The use of $H_2O$ reduces the energy yield of DBD by half. The efficiency of $H_2$ production is much better when microwave plasma and gaseous fuel are used. When gaseous fuels were injected into a microwave plasma in a metal cylinder, the $H_2$ output and energy yield were 0.18 kg/h and 0.043 kg/kWh, respectively, indicating effective energy transfer to $CO_2$ molecules. Moreover, $H_2$ has been recently produced using atmospheric 2.45 GHz plasmas with alcohol as a liquid fuel, albeit with low-energy yields [21–23]. However, the method developed in the present study was much more efficient, with corresponding values of 0.177 kg/h and 0.0591 kg/kWh, respectively, the efficiency of the reforming process can be maximized by using the waste heat of the plasma as a heat source for the catalyst. The hybrid system presented in this work represents a promising advance of CCU technology.

**Table 1.** Comparison of plasma-based methods of in terms of $H_2$ production rates and energy yields [24].

| Production Method | Initial Composition | Production Rate g($H_2$)/h | Energy Yield g($H_2$)/kWh | References |
|---|---|---|---|---|
| Liquid (vaporized) fuels | | | | |
| Water electrolysis | $H_2O$ | - | 20–40 | [25] |
| Gliding arc (water spray) | $H_2O + Ar$ | 0.004 | 13 | [21] |
| Dielectric barrier discharge | $CH_3OH + CO_2/H_2O$<br>$CH_3CH_2OH + CO_2$ | - | 3.3<br>6.7 | [22] |
| Microwave (2.45 GHz) plasma | $CH_3OH + Ar$<br>$C_2H_5OH + H_2O + Ar$ | 0.6<br>0.3 | 1.4<br>0.5 | [22] |
| Microwave (2.45 GHz) plasma | $C_2H_5OH + Ar$ | - | 0.55 | [23] |
| Microwave (2.45 GHz) plasma | $CH_3OH + Ar$<br>$C_2H_5OH + H_2O + Ar$ | - | 0.29<br>0.41 | [26] |

**Table 1.** *Cont.*

| Production Method | Initial Composition | Production Rate g(H$_2$)/h | Energy Yield g(H$_2$)/kWh | References |
|---|---|---|---|---|
| | | Gaseous fuels | | |
| Conventional steam reforming of methane (catalyst) | CH$_4$ + H$_2$O + Air | - | 60 | [25] |
| Electron beam radiolysis | CH$_4$ + H$_2$O | - | 3.6 | [27] |
| Dielectric barrier discharge | CH$_4$ + CO$_2$ | 0.25 | 5.2 | [28] |
| Dielectric barrier discharge | CH$_4$ + CO$_2$/H$_2$O | - | 0.5 | [29] |
| Metal-cylinder-based microwave plasma | CH$_4$ + CO$_2$/H$_2$O | 180 | 42.9 | [30] |
| Microwave (2.45 GHz) plasma | CH$_4$ + CO$_2$ | 240 | 41.4 | [19] |
| This work | CH$_4$ + CO$_2$ | 177 | 59.1 | - |

## 3. Experimental Setup

The experimental setup used to perform the experiments is presented in Figure 9a. The hybrid system was comprised of a microwave generator, a conversion reactor consisting of a microwave plasma torch and a microwave plasma-catalytic reactor), a gas supply, flow controller, an optical emission ultraviolet–near infrared (UV–NIR) spectrometer (Ocean Optics HR4000CG, Ocean Optics, Inc., Largo, FL, USA), and a gas analysis process. The main outlet gases synthesized by the microwave plasma-catalytic reactor included H$_2$O, CH$_4$, CO$_2$, CO, H$_2$, O$_2$, and other hydrocarbons. After H$_2$O was removed with a cold trap, the produced syngas was analyzed with a DCDA-2C-M gas flow meter (Sinagawa Corp., Tokyo, Japan). Additionally, gas analyzer (Advance Optima Continuous Gas Analyzers AO 2020, ABB Inc., Zurich, Switzerland) used in this work could detect CO (Uras 26, non-dispersive infrared gas sensor), H$_2$ (Caldos27, thermal conductivity), CO$_2$ (Uras 26, non-dispersive infrared gas sensor), CH$_4$ (Uras 26, non-dispersive infrared gas sensor), and O$_2$ (Magnos206, magneto-mechanical measuring). The optical emission spectrometer was used to determine the relative quantities of the species in the CO$_2$ microwave plasma. Emission spectra were recorded while employing a collimating lens attached to a tapered waveguide hole in the position closest to the center of the plasma where the emitted visible light was at its brightest.

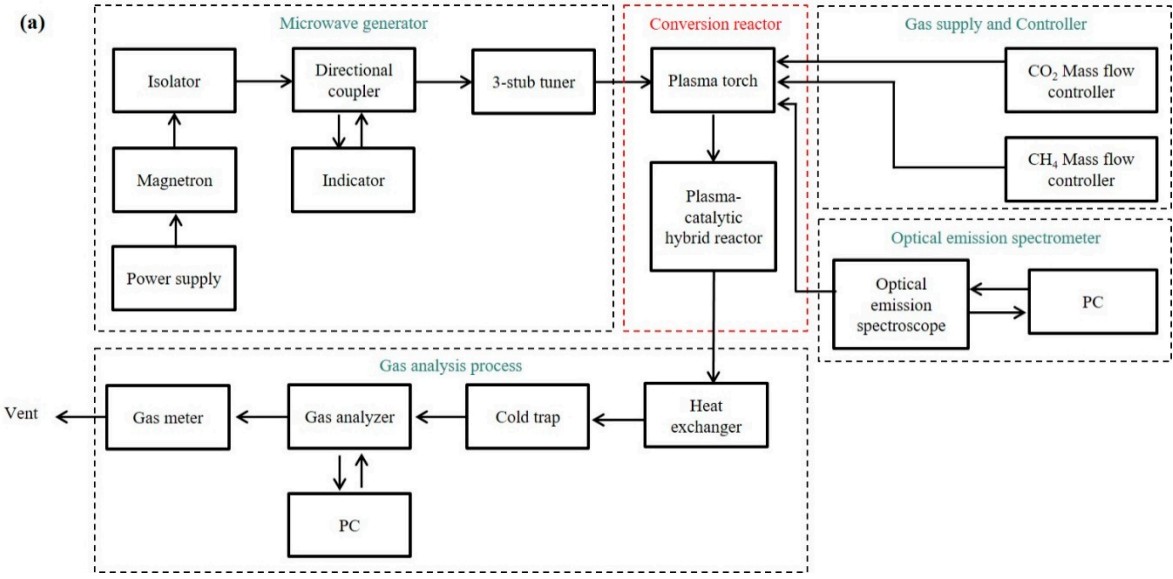

**Figure 9.** *Cont.*

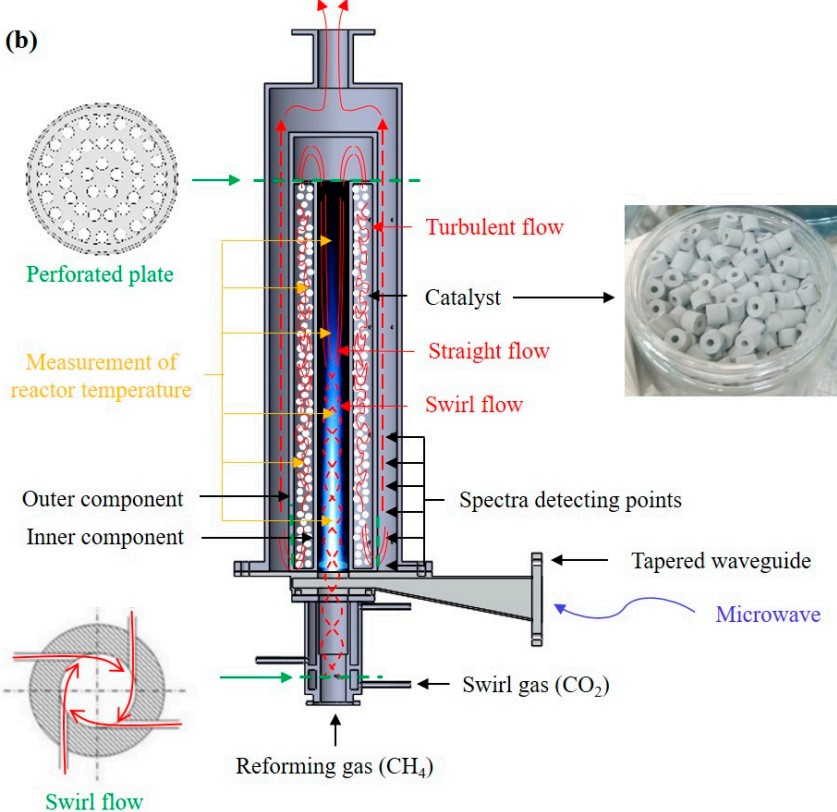

**Figure 9.** (**a**) Flow diagram and (**b**) schematic diagrams of the experimental setup of microwave plasma-catalytic reactor used for syngas production.

The image in Figure 9b shows the experimental setup for dry reforming from the microwave plasma-catalytic reactor. Details for the microwave plasma torch design in a hybrid system and its operation have been reported elsewhere [31,32]. The main component of the experimental configuration for efficient magnetron-to-gas microwave power transfer consisted of a magnetron head, an isolator, a directional coupler, and a three-stub tuner. Microwave power was transmitted from the magnetron head to the plasma torch via a WR340 waveguide, which had the potentially damaging effect of reflecting microwaves onto the magnetron. This was mitigated by the use of an isolator. The power induced by microwave radiation in the quartz tube was adjusted utilizing the three-stub tuner. Additionally, the reflected power was minimized to less than 1% of the forward power by adjusting the 3-stub tuner. A directional coupler was used to measure injected and reflected microwave power at the microwave plasma torch. The $CO_2$ microwave plasma torch was operated at a frequency of 2.45 GHz at atmospheric pressure, with the utilized power equaling 3 kW. The total gas flow rate was 30 L $min^{-1}$, with contribution of 15 L $min^{-1}$ from both $CO_2$ and $CH_4$. High-purity $CO_2$ (99.99%) was used for the microwave plasma to maintain a stable working plasma and to produce syngas via $CH_4$ conversion at atmospheric pressure. The quartz tube penetrated perpendicularly through the wide wall of the tapered waveguide. The dimensions of the quartz tube were 30 mm (outer diameter) $\times$ 2 mm (thickness) $\times$ 107 mm (length). The central axis of the quartz tube was located one-quarter wavelength from the short end of the tapered waveguide wall. The induced maximum electric field was located in the central axis of the quartz tube before the plasma was turned on. The $CO_2$ gas was injected into the microwave plasma torch via four small holes in tangential directions through a swirl holder which formed a swirl flow inside the quartz tube. $CH_4$ (99.99%) gas was also introduced as an additive into the plasma via the swirl holder. The $CO_2$ gas concentrated by swirl flow in the quartz tube stabilized the $CO_2$ microwave plasma in the form of a swirl within the quartz tube.

The microwave plasma-catalytic reactor was installed at the rear end of the plasma torch. The reactor consisted of a double component structure, which was divided into a plasma zone and a catalytic reaction zone. The microwave plasma flame was confined within the inner component of the reactor, and the outer component was filled with catalyst. Specifically, the quadralobe-shaped catalysts were packed irregularly in the outer compartment. The primary syngas was produced (swirl form) by microwave plasma torch was then passed through (straight flow) the inner component of the reactor. The primary syngas was next introduced into the outer component and underwent secondary reforming reaction (turbulent flow) with irregularly packed catalyst. The final syngas is produced after the catalytic process. Since primary syngas was introduced into the catalytic reaction zone after the plasma treatment step, the catalytic process of the hybrid reactor can considerably increase reforming efficiency. The high temperature of the microwave plasma flame supply heat to the reactor wall between the plasma reaction zone and the catalytic reaction zone. Additionally, the gas reformed by the plasma maintains the temperature of the catalytic reaction zone and increases the reforming efficiency.

The catalyst used for dry reforming was a commercial nickel-based catalyst (HyProGen R-70, CLARIANT, Basel, Schweiz), which provided high coking resistance, robust physical integrity and a long operating life. Ring type catalysts were used in this experiment. The pellets were $8 \times 8 \times 3$ mm in size, and the ring shape provided a large surface area. The catalysts operate across broad temperature range of 400–900 °C. The catalysts were used to fill in the outer component of the microwave plasma-catalyst reactor for the experiment.

The reforming gas–catalyst reaction time was estimated using the parameter of gas hourly space velocity (GHSV), with GHSV itself calculated as:

$$\text{GHSV} \left(\text{h}^{-1}\right) = \frac{\text{flow rate } \left(\text{m}^3\,\text{h}^{-1}\right)}{\text{catalyst volume } \left(\text{m}^3\right)} \tag{7}$$

Using the volume of the quadralobe-shaped catalyst, we calculated the GHSV to be 5160 $\text{h}^{-1}$. The flow rate of the primary syngas after the plasma process was 43 L $\text{min}^{-1}$, and the catalyst volume used for the reforming process was 0.5 L. Since the reforming efficiency is influenced by the residence time in the catalyst bed, the amount of used catalyst decreases with increasing space velocity. In addition, the conversion efficiency of the hybrid system was also calculated that the plasma waste heat was used for the catalytic reaction.

A small amount of carbon powder was produced along with the primary syngas by the microwave plasma. Carbon powder is produced because carbon dioxide is not completely converted to syngas. However, the amount was negligible. The carbon solids on the inner wall of the reactor were immediately burned in the high-temperature microwave-driven region, and the remaining carbon solids were blown out through the exit from the plasma zone. The hybrid system also included a carbon powder trap that was arranged at the gas exhaust end of the plasma reforming zone. This is the tap for removing carbon particulates, which were not reacted in the reforming reactor.

## 4. Conclusions

Herein, we successfully applied a $CO_2$ microwave plasma–catalytic reactor at a frequency of 2.45 GHz at atmospheric pressure to the production of syngas via $CH_4$ conversion. We demonstrated that the hybrid system was well suited for $CO_2$ splitting and could be operated at high conversions. Analysis of the $CO_2$ microwave plasma by optical spectroscopy indicated that it provided sufficient oxygen radicals for dry reforming at a high temperature of 6760 K at atmospheric pressure. Although inferior conversions were observed with the non-catalyzed plasma process at low power, $CO_2$ and $CH_4$ could be almost completely converted into syngas under hybrid system conditions. The optimized conversion ratios of $CO_2$ and $CH_4$ were 69.3% and 81.2%, respectively, and optimized $H_2$ and CO mass yield rates of 0.177 kg/h (59.1 g/kWh) and 1.58 kg/h, (527 g/kWh), respectively were measured.

Importantly, the proposed hybrid system provided a low cost method with good efficiency of 62.1%. Thus, we have demonstrated that the developed hybrid system is well suited for $CO_2$ reduction and syngas production. Furthermore, the syngas produced by the hybrid system can be used not only for the production of acetic acid or methyl formate; it also yields the $H_2$:CO mole ratio required for the production of various substances in combination with wet syngas processes. $CO_2$ was employed as the swirl gas at a flow rate of 150 L min$^{-1}$ in a recent experiment. The microwave plasma had a diameter and length of 6.5 cm and 100 cm, respectively, at an applied power of 30 kW at 915 MHz. However, the diameter of the $CO_2$ microwave plasma flame produced in this experiment was ~2.6 cm. Compared to the 2.45 GHz torch plasma, the 915 MHz plasma source generated an enlarged plasma based on the waveguide, which considerably increased the plasma volume. If the large plasma volume and the high temperature generated by the 915 MHz microwave source can be used efficiently, large amounts of $CO_2$ and $CH_4$ can be converted and the reforming efficiency can be increased. Finally, this technology can be used in industries that are the main sources of landfill and $CO_2$ gas. Furthermore, because the carbon powders produced in the reforming reaction deactivate the catalysts, further research to remove carbon deposition problems on the catalysts is required.

**Author Contributions:** S.M.C. performed the literature search and drafted the manuscript; Y.C.H. made great efforts to revise the manuscript; D.H.S., S.H.M., and G.W.Y. participated in the checking.

**Funding:** This research was funded by "the 2013 R&D Convergence Program" of the Korea Research Council of Fundamental Science and Technology (Convergence-13-5-NFRI); and "the R&D Program of the Plasma Convergence & Fundamental Research" through the National Fusion Research Institute of Korea (NFRI) funded by the Korea Government.

**Conflicts of Interest:** The authors declare no conflict of interest.

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
