# Peer review of "CO2 Microwave Plasma—Catalytic Reactor for Efficient Reforming of Methane to Syngas"

_catalysts, doi:10.3390/catal9030292_

Round 1

Reviewer 1 Report

see attachment

Author Response

Response to Reviewer 1 Comments

Review of “Microwave Plasma-Catalytic Hybrid Reactor for Efficient Carbon Dioxide Utilization”

By S.M. Chun et al, submitted to Catalysis

This study reports high conversions of CH4 and CO2 to syngas (around 90%) with high thermal efficiency (86%) using CO2 microwave plasma in a first stage conversion with Ni on calcium aluminate thermos-catalytic reforming in a second stage using the residual heat from the CO2 plasma. 

The reported experiments are of high quality, making use of appropriate methods for plasma diagnostic of composition (many radicals C and O based radicals by Optical Emission UV-NIR Spectrometry) and plasma temperature (ca. 6760 K) by fitting of the rotational temperature of C2 swan band, also an output of OES. 

The experimental set up is impressive in particular the generation of the CO2 plasma, the size of the reactor (3 kW plasma flame, as stable plasmas are reported to require minimum flows to operate, and 500 g of catalyst).  The manuscript is worth publishing but a few minor aspect would benefit from being addressed, outlined below:

Point 1: The title is too vague, CO2 utilization can be effected in many ways, this paper focusses on dry reforming (reaction CH4+CO2 = 2CO + 2H2), MW-CO2 plasma-catalytic reactor for efficient reforming of CH4 to syngas would be more informative.

Response 1: In response to the reviewer’s comment, the title of manuscript was revised to “CO2 Microwave Plasma - Catalytic Reactor for Efficient Reforming of Methane to Syngas

Point 2: The insert in figure 1 depicting the experimental reactor is too small to read already at A4 size, it is advised the captions in this part of the figure be enlarged so that the figure is ultimately legible. It is a very important figure because without it is difficult to understand where the reactant feeds are positioned with respect to the plasma, and where the catalyst is located.

Response 2: In accordance with the reviewer’s comment, we enlarged the next in Fig. 1 (a) and (b) on page 3.

Point 3: Description of the catalyst is confusing, it is described as honeycomb monolith, and Ni on calcium aluminate. In figure 1 it is difficult to reconcile the honeycomb monolith structure with a cylindric wall surface coverage, honeycomb monoliths are normally used to covers the whole cross section area of the flow, not just what appears as just an annular thin layer near the reactor cylindrical wall, this would not make best use of their surface area or residence time.

Response 3: The description of the catalyst-shape in the original manuscript contained an error. The shape of the catalyst used in this experiment was a domed cylinder-type quadralobe with four holes and 4 flutes. The pellets were 10.5mm x 13mm in size, and the quadralobe shape provided a large surface area. The catalyst of quadrilobe shape can provide a high-surface area. The catalysts were used to fill in the outer component of the microwave plasma-catalyst reactor for the experiment. We removed the sentence mentioned by the reviewer and added three sentences on page 5 to describe the catalyst.

Point 4: Another question is, does Ni on calcium aluminate represent the washcoat of the monolith? If so, what was the wt% of Ni the calcium aluminate or as a wt% of monolith?

Response 4: The catalyst consisted of 60% nickel oxide and 40% aluminum oxide. In response to the reviewer’s comment, we corrected the sentence on page 5 to accurately describe the catalyst composition.

Point 5: Since this appears to be the first publication with this particular set up, it would help to have figure zoom-in of what happens to the flow near the catalyst, does it convert from axial in the plasma flame to outer radial through the honeycomb and back to annular after the honeycomb to join up at an axial top exit?

Response 5: We removed the sentence to which the reviewer referred and added six sentences on page 5 to explain the catalytic hybrid reactor. The microwave plasma-catalytic reactor consisted of a double component structure, which was divided into a plasma zone and a catalytic reaction zone. The microwave plasma flame was confined within the inner component of the reactor, and the outer component was filled with catalyst. Specifically, the quadralobe-shaped catalysts were packed irregularly in the outer compartment. The primary syngas was produced (swirl form) by microwave plasma torch and passed through (straight flow) the inner component of the reactor. Next, the primary syngas was introduced into the outer component and underwent secondary reforming reaction (turbulent flow) with irregularly packed catalyst.

Point 6: When the GHSV is calculated, what is used as catalyst volume, that of the washcoat, or the honeycomb structure?

Response 6: We calculated the GHSV to be 5160 h-1 using the volume of the quadralobe-shaped catalyst. The flow rate of the primary syngas after the plasma process was 43 L/min, and the catalyst volume used for the reforming process was 0.5 L. We added two sentences on page 5 to explain the GHSV.

Point 7: This paper is submitted to Catalyst, so information about the catalyst’s morphology and composition is important for future studies on this type of plasma-catalytic process. CCU is a hot topic and this paper could open a bright future for CO2 chemical conversion, the paper could be cited many times if information on the catalyst was more forthcoming. ABB’s analyser Uras 26 does not measure H2 or O2, Uras 26 uses NDIR absorption to measure CO, CH4 and CO2., and other analysers need to be used for H2 (eg. TCD, with ABB’s Caldos), and O2 (e.g. paramagnetic, with ABB’s Magnos), it is surprising to see in the text that a Uras 26 from ABB measures the whole set CH4, CO2, CO H2 and O2.

Response 7: The Uras26, Caldos27, and Magnos206 analyzer were used for gas analysis, but two of the gas analyzers were not mentioned in the original manuscript. To address the reviewer’s comment, we added the Caldos27 and Magnos206 analyzers on page 3.

Point 8: There is a confusing concept described at the bottom of page 5 which describes synthesis gas being introduced into the catalytic reaction area after the plasma treatment step. Since synthesis gas is the ultimate product of the whole plasma-catalytic hybrid, why would it be introduced again before the plasma, it sounds like ‘products recirculation’ and this would be adverse to the both the kinetics and thermodynamics of the CH4+ CO2 conversion to 2CO+2H2 by decreasing both reactants partial pressure. After reading the whole paper, the reader realizes with hindsight that this ‘synthesis gas’ introduction into the catalytic reaction area is just partially converted gas flow by the plasma alone, which according to fig.6 had achieved just 40% CH4 conversion and 30 % CO2 conversion. The authors need to clarify that this is not syngas per se, it is the product gas of the plasma alone stage. At the bottom of page 6 the term primary syngas is used, this term could be introduced earlier by explaining there is primary (after plasma) and final (after catalyst) syngas.

Response 8: To clarify the distinction pointed out by the reviewer, we refer to the gas following the plasma process as the “primary syngas” on page 4, while the gas following exposure to the catalyst is referred to as “final syngas”.

Point 9: The role of the wet carbon –refining device is unclear, what does it do exactly? Trap particulates of char or carbon? Remove condensable flammable gases? Please use more concrete words than ‘refining’ carbon in the primary syngas

Response 9: We added the following sentences on page 5 to address the reviewer’s comments: “A small amount of carbon powder was produced along with the primary syngas by the microwave plasma. Carbon powder is produced because carbon dioxide is not completely converted to syngas, but the amount was negligible. The carbon solids on the inner wall reactor were immediately burned in the high-temperature microwave-driven region, and the remaining carbon solids were blown out through the exit from the plasma zone. The hybrid system also included a carbon powder trap that was arranged at the gas exhaust end of the plasma reforming zone. It is the tap for removing carbon particulates, which is not reacted in the reforming reactor.” Additionally, we removed the phrase “wet carbon-refining device”.

Point 10: Section 3.1 is overlong and in a large early portion speculative, the authors put ahead a discussion of the reaction CO2 ->CO + ½ O2 which is meant to occur via CO2->CO+O. However whereas there is evidence of radicals O being generated in the CO2 plasma (fig. 3a, not discussed at this stage), there is no evidence of O2 every being produced or detected, personally I find hard to swallow that CO2 plasma can be used as an stable oxygen generator, because the O radicals, once produced at very high temperatures, would much prefer reacting with anything else than themselves, so I recommend that this whole chunk of discussion of around the O2 producing reaction is scrapped. Much of section 2.1 can be shortened by keeping the right references and conveying the fact that at the CO2 plasma torch temperatures, with CH4 feed, pretty much a soup of highly reactive radicals are obtained resulting in the cracking of both molecules and recombination into syngas, basically from page 8 onwards.

Response 10: As recommended by the reviewer, we removed the entire discussion regarding the O2 producing reaction to avoid confusing the readers.

Point 11: One figure appears to have lost its caption and is currently below figure 4 on page 13. Shouldn’t this be after figure 1 and the other figures re-numbered?

Response 11: The figure mentioned by the reviewer is Fig. 5(a). In response to the reviewer’s comment, we placed of the figure in the correct location.

Point 12: Figure 4 shows the temperature simulation of the rotational temperature of the C2 swan emission, it would help in its discussion to provide a value to an acceptable range of goodness of fits and how figure 4 is within this range, as all rotational temperature spectra, this one is very noisy, and it is hard to visually check whether the fit is good enough, very good or poor. Comparison with values of goodness of fit from the literature would be welcome. I understand that when you are in regions of several thousands of K, it may not matter much we have plus or minus hundreds of K but the readership of Catalyst may not be familiar with temperature measurements in plasmas.

Response 12: We add four sentences on page 12 to explain the figure 4. Fig. 4 has been re-number to Fig. 3(c) to focus on plasma characterization using OES. The experimental data and the simulated spectrum in Fig. 3(c) appear noisy and do not completely overlap. However, overlaying the experimental data and the simulated spectrum yielded good results, and the error of the fit was ±300 K of the plasma temperature. The figure indicates that the temperature of the CO2 microwave plasma itself was sufficient for the reforming reaction. Descriptions of the catalyst and the plasma are clearly divided into sections.

Point 13: In figure 5 the legend shows temperature profiles ‘outside’ and ‘inside’, F is misspelled to inutside, but aside from that the concepts of what represent inside temperatures and outside temperatures are not well defined, the text should, helped with reference to a proper figure indicate where these are measured. The term internal temperature is also used in page 16, is also confusing, since it seems to be around 650 C when we know the flame reaches +6000K. 

Response 13: We corrected the meaning of temperature in section 3.3 and the typo in Figure 5(b) to clarify the issue pointed out by the reviewer.

Point 14: Page 17 refers again to molecular O2, this time, excited for the reaction C+O2*->CO2, when there is no evidence of O2 being detected anywhere, excited or otherwise.

Response 14: We removed the sentence mentioned by the reviewer.

Point 15: Figure 8 needs the word ‘energy’ placed before the efficiency y axis label. 

Response 15: We corrected the y-axis label in Fig. 8 to “Energy efficiency (%)”.

Point 16: This reviewer recommends for the sake of those who will read the paper in black and white or grey shades to choose a clear colour (or textured fills) for the plasma alone and a dark colour for plasma + catalyst throughout the paper, because they currently look the same when colour is removed.

Response 16: Based on the recommendation from the reviewer, we added different shades and textures to make is easier to distinguish between data from the plasma alone and data from the plasma + catalyst.

Point 17: Equations 6 and 7 on page 17 provide the conversions of CO2 and of CH4 with the [CO2]in, [CO2]out, [CH4]in and [CH4]out terms without provision of units. Usually the [XX] symbol refers to concentration, e.g. in mol/cm3. In the case of conversion calculation this would be wrong, conversion should be based molar flows (e.g. mol/s), the difference is important because the CO2 reforming of CH4 is non equimolar, it generates twice as many moles of product that reactant. In order to determine molar flows out, one needs the mol fraction (provided by the ABB analysers) and the total molar flow out. The latter can only be determined via direct flow measurement or by elemental balance (e.g. of carbon ). The authors need to reassure the reader the correct conversion calculation was used (based on flows, not concentrations) and by which method they obtained the molar flows.

Response 17: Figure 5 shows three results, including (a) concentration, (b) flow rate, and (c) conversion (%) after the experiment. The total gas flow rate in the experiment was 30 L min–1 (15 L min–1 of CO2 and 15 L min–1 CH4). However, catalyst addition yield 1.746 times more syngas than reactants, based on the flow rates measured after exposure to the catalyst. Fig. 5 (b) shows that catalyst addition increased the flow rates of H2 and CO from 11.91 to 25.54 L min–1 and from 10.71 to 26.84 L min–1, respectively. Therefore, the results shown in Fig. 5 (b) were applied in equations (5) and (6) to calculate conversion of CO2 and CH4. We added six sentences, and corrected equations (5) and (6) in accordance with reviewer’s comment.

Point 18: The paper is generally well written, but a few typos and poor grammar slipped into pages 19 and 20.

Response 18: We corrected the typos and grammatical errors on page 17.

Reviewer 2 Report

This paper is a very interesting addition to the field of carbon dioxide-based chemicals, in particular to the classical reforming process to methanol. The manuscript shows important results on the used of microwave plasma coupled to classical catalysts for the production of hydrogen, carbon monoxide and dioxide out of methane for further use to obtain other chemicals such as methanol, showing the superior performance of a hybrid system that combines MW-plasma and catalysts.

My minor comments and corrections are on the attached pdf file (just a few corrections to typos and use of english).

On the other hand, I wonder how this hibrid technology compares in terms of scalability to other well-known dry and vapor-based reforming processes for methane. Scale-up is scarcely addressed in the manuscript and would be convenient to add some comments on it.

Author Response

Response to Reviewer 2 Comments

This paper is a very interesting addition to the field of carbon dioxide-based chemicals, in particular to the classical reforming process to methanol. The manuscript shows important results on the use of microwave plasma coupled to classical catalysts for the production of hydrogen, carbon monoxide and dioxide out of methane for further use to obtain other chemicals such as methanol, showing the superior performance of a hybrid system that combines MW-plasma and catalysts.

Point 1: My minor comments and corrections are on the attached pdf file (just a few corrections to typos and use of english).

Response 1: We corrected the typos and grammatical errors in the manuscript.

Point 2: On the other hand, I wonder how this hybrid technology compares in terms of scalability to other well-known dry and vapor-based reforming processes for methane. Scale-up is scarcely addressed in the manuscript and would be convenient to add some comments on it.

Response 2: The microwave plasma source of the hybrid system can greatly improve the plasma volume using the frequency modulation. This means that single system can be applied to large scale reforming process. In a recent experiment, using CO2 as the swirl gas at a flow rate of 150 L min–1, the microwave plasma had a diameter and length of 6.5 and 100 cm, respectively, at an applied power of 30 kW. On the other hand, the diameter of the CO2 microwave plasma flame produced this experiment was about 2.6 cm. In comparison with the 2.45 GHz torch plasma, the 915 MHz plasma source had an enlarged plasma based on the waveguide, which considerably increased the plasma volume. Therefore, if the large volume of plasma and the high temperature generated by the 915 MHz microwave source could be used efficiently large amounts of CO2 and CH4 could be converted, and the reforming efficiency could be increased. We are currently constructing a larger and more efficient catalyst-integrated microwave plasma torch operating at 915 MHz. In addition, we added content related to scale-up on pages 18 and 19 in accordance with the reviewer’s comment.

Reviewer 3 Report

This manuscript described the application of microwave system in reforming of methane to hydrogen and related gases. The design of developed system was thoroughly described and its operation and efficiency were reported. However, the discussion on the preparation and characterizations of catalysts was missing. Since “Catalysts” is a scientific journal in the field of catalysis, I strongly suggest the authors to add this part to meet the requirement of Journal. Some comments are listed below.

1.      Please make sure that all the significant figures in the manuscript and SI are meaningful.

2.      Please define all the acronyms when they appears in the text for the first time.

3.      An overall review of literature is suggested. The papers cited in the manuscript are old.

4.      The efficiency of developed system (microwave plasma-catalytic hybrid reactor) was discussed. However, please comment on the loss of efficiency if the developed system is utilized in the industry, such as electric plant, and compare to the conventional process.

5.      The technology for separation of mixture gases (CO2/CO/CH4/H2) in the developed system can be commended.

6.      The preparation and characterizations of catalysts are quite superficial. I think that it is necessary to report the detail of catalysts if the authors would like to submit a paper in the field of catalysis, such as Catalysts.

Author Response

Response to Reviewer 3 Comments

This manuscript described the application of microwave system in reforming of methane to hydrogen and related gases. The design of developed system was thoroughly described and its operation and efficiency were reported. However, the discussion on the preparation and characterizations of catalysts was missing. Since “Catalysts” is a scientific journal in the field of catalysis, I strongly suggest the authors to add this part to meet the requirement of Journal. Some comments are listed below.

Point 1: Please make sure that all the significant figures in the manuscript and SI are meaningful.

Response 1: We have corrected the figures and the content of the manuscript in accordance with the reviewer’s comment.

Point 2: Please define all the acronyms when they appears in the text for the first time.

Response 2: In accordance with the reviewer’s recommendation, we have defined the acronyms used in manuscript.

Point 3: An overall review of literature is suggested. The papers cited in the manuscript are old.

Response 3: We deleted the References 14, 15, 16, and 18 mentioned by the reviewer. Additionally, we added four references on page 19 and 20.

Point 4: The efficiency of developed system (microwave plasma-catalytic hybrid reactor) was discussed. However, please comment on the loss of efficiency if the developed system is utilized in the industry, such as electric plant, and compare to the conventional process.

Response 4: Compared to plasma-based and catalytic technologies developed over the past few years, this system (a microwave plasma-catalytic hybrid reactor) is highly efficient. However, it is difficult to assess the efficiency of this system for application in an industrial setting, such as a power plant. Pure CO2 and CH4 (99.99%) were used in this experiment for dry reforming. The storage gas of a CCS process in a power plant may contain some impurities, such as NOx and SOx. In order to apply this system in a power plant, we need to know the purity of and impurities in the storage gas. In addition, further research is needed for economic evaluation and scale-up of this system. In other words, the issues mentioned are considered factors that could reduce the efficiency of this system.

Point 5: The technology for separation of mixture gases (CO2/CO/CH4/H2) in the developed system can be commended.

Response 5: The syngas produced by this system may contain unreacted gas, such as CO2 and CH4. However, the syngas can be converted into high-value products, such as methanol or synthetic fuels, via the Fischer Tropsch (FT) process. To be used in these processes, the syngas composition must be adjusted in terms of ideal H2/CO ratios. Amine absorption is currently used for syngas composition adjustment, but PSA processes have been investigated as viable alternatives. In addition to the design of new processes, novel adsorbent materials that can improve performance have recently become a focus of research. On the other hand, the main goal of this experiment was to explore effective methods for production of syngas or hydrogen. We do appreciate the reviewer’s comment, although the technology of syngas separation is not our field of research.

Point 6: The preparation and characterizations of catalysts are quite superficial. I think that it is necessary to report the detail of catalysts if the authors would like to submit a paper in the field of catalysis, such as Catalysts.

Response 6: To address the reviewer’s comment, we have added a more detailed explanation of the preparation and characterization of the catalyst to page 5.

Round 2

Reviewer 3 Report

The revised manuscript has been improved by adding the information of the catalyst. However, the discussion on the fresh and used catalysts is missing. I strongly suggest the authors to examine the change in structural properties of fresh and used catalysts, at least the XRD pattern, nitrogen adsorption-desorption isotherm. Besides, the catalytically active sites can be  proposed.

Author Response

Point 1: The revised manuscript has been improved by adding the information of the catalyst. However, the discussion on the fresh and used catalysts is missing. I strongly suggest the authors to examine the change in structural properties of fresh and used catalysts, at least the XRD pattern, nitrogen adsorption-desorption isotherm. Besides, the catalytically active sites can be proposed.

Response 1: The structure characteristics of the fresh and spent catalysts were analyzed by adsorption-desorption method and X-ray diffraction. We added content related to structure properties of fresh and used catalyst on pages 13, 14 and 20 in accordance with the reviewer’s comment
